# Demography and the emergence of universal patterns in urban systems

Luís M. A. Bettencourt[1,2,3] & Daniel Zünd [1,2✉]

Urban areas exist in a wide variety of population sizes, from small towns to huge megacities. No proposed form for the statistical distribution of city sizes has received more attention than Zipf's law, a Pareto distribution with power law exponent equal to one. However, this distribution is typically violated by empirical evidence for small and large cities. Moreover, no theory presently exists to derive city size distributions from fundamental demographic choices while also explaining consistent variations. Here we develop a comprehensive framework based on demography to show how the structure of migration flows between cities, together with the differential magnitude of their vital rates, determine a variety of city size distributions. This approach provides a powerful mathematical methodology for deriving Zipf's law as well as other size distributions under specific conditions, and to resolve puzzles associated with their deviations in terms of concepts of choice, symmetry, information, and selection.

[1] Mansueto Institute for Urban Innovation, The University of Chicago, Chicago, IL, USA. [2] Ecology & Evolution, Sociology, The University of Chicago, Chicago, IL, USA. [3] Santa Fe Institute, Chicago, IL, USA. ✉email: dzuend@uchicago.edu

The observation of approximate power law distributions of type frequencies is very common in complex systems[1–5]. The emergence of such distributions is known to be intimately connected to multiplicative growth processes mediated by the structure of complex networks of interaction[6,7]. Many different effective models have been proposed to derive particular forms of these distributions. They share common characteristics, such as invoking proportional growth (also known as Gibrat's Law) or preferential attachment and starting out at some mesoscopic scale, typically not derived from the fundamental population dynamics of the system[7–18].

Among the many statistical patterns that characterize size distributions, none has received more attention than Zipf's law[17–21]. Specifically, Zipf's law applied to an urban system states that the expected population size of a city $N$ is a function of its size rank, $r$,

$$N(r) = \frac{N_0}{r^z}, \qquad (1)$$

with an exponent $z = 1$, where $N_0$ is the size of the largest city ($r = 1$). In other words, the largest city is predicted to be twice the size of the second largest city ($r = 2$), three times larger than the third largest ($r = 3$), and so on. This is equivalent to a Pareto probability density for city sizes, $P_z(N) = c/N^{1+z}$, with $c$ a normalization constant. First observed by Auerbach for city sizes in 1913[22], Zipf's law gained widespread attention through the work of the linguist George Kingsley Zipf, who established the rank-size rule first in the context of word frequencies in the 1930s[23] and then cities[24]. Zipf's law quantifies the concept of urban hierarchy, which, together with ideas of central place, location theory and other "laws" characterizing statistical regularities in urban systems, became the basis for the quantitative revolution in geography in the decades after World War II. Systematic analyses of urban systems from that time already suggested that the city size distribution is not universal, acquiring different forms as a result of distinct historical patterns of growth that do not have a relation to levels of economic performance or human development[25].

Like other power laws in complex systems, Zipf's law has also attracted much scrutiny and criticism over time[6,17]. It is hardly ever observed empirically in an unambiguous way, with data often showing systematic deviations at the two extremes of very large and especially of very small cities[26–34]. Besides such systematic variations, estimates of the exponent $z$ take substantial ranges across time and for different urban systems, for example in China[27,35,36], the United States[20,21,37], and European countries[9,38].

In our view, the empirical controversies about the form of the city size distribution and the interpretation of associated deviations can only be resolved through more fundamental approaches that start out with a system's basic population dynamics. It is also in the context of population dynamics that we can make sense of processes of selection and randomization that order and disorder complex systems. In this paper, we adopt the lens of evolutionary population dynamics to show the emergence and meaning of Zipf's distribution as a neutral law, signaling the absence of selection. It follows that associated deviations acquire the meaning of information, specifying people's preferences about where to live, constitute a family, and die. These results allow us to discuss and resolve a number of puzzles surrounding the interpretation of Zipf's law as a signal for urban system integration and associated implications for the coherence and symmetry of population states.

## Results

### Demographic dynamics of urban systems
We now derive the population size distribution of cities in the most general way, set by their demographic dynamics. The change of the population $N_i$ in city $i$ during unit time $\Delta t$ necessarily results from the balance of births, deaths, and migration as

$$N_i(t + \Delta t) = N_i(t) + \Delta t \left[ v_i N_i(t) + \sum_{j=1, j \neq i}^{N_c} (J_{ji} - J_{ij}) \right], \qquad (2)$$

where $N_c$ is the total number of cities, $v_i = b_i - d_i$ is the city's vital rate, the difference between its per capita average birth rate, $b_i$, and the death rate, $d_i$. The current $J_{ij}$ is the number of people moving from city $i$ to city $j$ over the time interval $\Delta t$, and vice-versa for $J_{ji}$. For simplicity of notation we will work in units where $\Delta t = 1$. Eq. (2) only accounts explicitly for migration between different specific cities, which can cross political borders. Migration to city $i$ from non-specific places outside the system, for example from non-urban regions or other nations, can be incorporated into the vital rates $v_i$. Note that this model is very general, and naturally accommodates the inclusion of new cities over time when their sizes become non-zero, and indeed the disappearance of others, if their size vanishes.

To proceed we need to explore the dependence of the currents $J_{ij}$ on population size. We start by writing these currents as population rates, $J_{ij} = m_{ij} N_i(t)$, where $m_{ij}$ is the probability that someone in city $i$ moves to city $j$ over the time period. This allows us to write Eq. (2) in matrix form

$$\mathbf{N}(t) = \mathbf{A}(t)\mathbf{N}(t - 1), \qquad (3)$$

where $\mathbf{N}(t) = [N_1(t), \ldots, N_i(t), \ldots, N_{N_c}(t)]^T$ is the vector of populations in each city at time $t$ and the matrix $\mathbf{A}$ projects the population at time $t - 1$ to time $t$. This matrix is known in population dynamics as the environment[39], with elements

$$A_{ij} = \begin{cases} 1 + v_i - m_i^{\text{out}}, & \text{if } i = j \\ m_{ji}, & \text{if } i \neq j \end{cases}, \qquad (4)$$

where $m_i^{\text{out}} = \sum_{k=1, k \neq i}^{N_c} m_{ik}$ is the total probability that a person migrates out of city $i$ per unit time. We also define the population structure vector $\mathbf{x} = [x_i]$, with $x_i = N_i/N_T$, which is the probability of finding a person in city $i$ out of the total population $N_T$. Note that $\sum_{i=1}^{N_c} x_i = 1$, so that the structure vector has $N_c - 1$ degrees of freedom (the magnitude of $x$ is fixed).

Equation (3) can now be solved by repeated iteration

$$\mathbf{N}(t) = \mathbf{A}(t)\mathbf{A}(t - 1) \ldots \mathbf{A}(1)\mathbf{N}(0). \qquad (5)$$

**City size distributions in different environments.** A number of important ergodic theorems[40] in population dynamics tell us about the properties of the solutions under different conditions on the environments $\mathbf{A}(t)$. These results rely on some constraints on the properties of $\mathbf{A}$; specifically, that the product of matrices in Eq. (5) is positive for sufficiently long times[39,41].

First, the weak ergodic theorem guarantees that for a dynamical sequence of environments, the difference between two different initial population structure vectors decays to zero over time. This means that there is typically an asymptotic city size "distribution", which is a function of environmental dynamics only, independent of initial conditions. When the environment is stochastic but otherwise time independent, the strong stochastic ergodic theorem states that the structure vector becomes a random variable whose probability distribution converges to a fixed stationary distribution, regardless of initial conditions. This is the sense in which most derivations of Zipf's law apply[17,33]. For stochastic environments, only probability distributions of structure vectors, not vectors themselves, can be predicted. Finally, in situations where the environment is time dependent and stochastic, the weak stochastic ergodic theorem states that the difference between the probability

distributions for the structure vector resulting from any two initial populations, exposed to independent sample paths, decays to zero. Again, in cases where the environment is explicitly dynamic, besides being stochastic in a stationary sense, we cannot say much about the actual probability distribution of city sizes, only that the importance of initial conditions vanishes for sufficiently long times.

To get more intuition on these results we will now show some explicit solutions. The city size distribution can be calculated exactly when the environments **A** are arbitrarily complicated, but static. This is the result of the strong ergodic theorem, which says that in a constant environment **A**, any initial population vector converges to a fixed stable structure given by the leading eigenvector of the environment[39]. This is guaranteed to exist if **A** is a strongly connected aperiodic graph, meaning that any city can be reached from any other city in a finite and diverse number of intermediate steps along the non-zero migration flows between them. This is a reasonable assumption to make about an urban system, which may even serve as a definition. The leading eigenvector of **A** is the eigenvector centrality of the urban system. It describes the probabilistic location of a random walker over the graph of migration flows[42]. This is equal to the stationary solution of numerically integrating a network described by environment **A**. Figure 1 shows two numerical solutions with different initial conditions, but the same environment **A**. The cities in the two runs converge to similar sizes, as they experience a common environmental matrix with all entries set to be the same plus a little noise. The insets show how the population structure converges from both initial conditions to the one defined by the leading eigenvector $\mathbf{e}_0$. Because the structure vector is the probability of finding a person out of the total population in the urban system in a specific city, we use the Kullback-Leibler (KL) divergence[43] $D_{KL}(P|P_{\mathbf{e}_0}) = \sum_x P[x]\log P[x]/P_{\mathbf{e}_0}[x]$ as a measure of the 'distance' between the initial distribution, $P$, and the final, $P_{\mathbf{e}_0}$. The KL divergence is a foundational quantity in information theory. It measures the information gain in describing a probabilistic state using one distribution, $P_{\mathbf{e}_0}$, versus another, $P$, in units of information (bits). Note that $P \to P_{\mathbf{e}_0}$ and thus $D_{KL}(P|P_{\mathbf{e}_0}) \to 0$ over time, see insets in Fig. 1.

The convergence rate of the city size distribution to the leading eigenvector is given by the ratio of secondary eigenvalues to the dominant one. Explicitly, we can now write this solution as

$$\mathbf{N}(t) = \mathbf{A}^t\mathbf{N}(0) \to \mathbf{N}(t) = \sum_{i=1}^{N_c} (\lambda_i)^t c_i\mathbf{e}_i. \qquad (6)$$

where $\mathbf{Ae}_i = \lambda_i\mathbf{e}_i$, so that $\mathbf{e}_i$ are the eigenvectors of the matrix **A** and $\lambda_i$ the corresponding eigenvalues, so that $\lambda_0 > Re(\lambda_1) > Re(\lambda_2) > \ldots$, where $Re(\lambda_i)$ is the real part of the eigenvalue. The projection coefficients $c_i$ are such that $\mathbf{N}(0) = \sum_{i=1}^{N_c} c_i\mathbf{e}_i$, which can be written as $\mathbf{c} = \mathbf{E}^{-1}\mathbf{N}(0)$, where **E** is a matrix whose $j$th column vector is $\mathbf{e}_j$. The strong ergodic theorem states that the largest eigenvalue $\lambda_0$ is positive and real and that the associated eigenvector $\mathbf{e}_0$ is also positive in terms of all its entries. This means that, over time, the dynamics of the urban system approaches that of the growth of its dominant eigenvector, because the projections on all other eigenvectors decay exponentially in relative terms, as

$$\left(\frac{\lambda_i}{\lambda_0}\right)^t = e^{-\ln\frac{\lambda_0}{\lambda_i}t} \xrightarrow[t\to\infty]{} 0 \qquad (7)$$

and exemplified in the two insets in Fig. 1. The longest characteristic time is associated with the difference in magnitude between the two largest eigenvalues $t_* = 1/\ln\frac{\lambda_0}{|\lambda_1|}$. Consequently, for $t \gg t_*$ we observe an extreme dimensional reduction from an initial condition characterized in general by $N_c$ degrees of freedom to a final one with just one, set by the environment's leading eigenvector. Using the values of migration flows in the US urban system over the last decade, we can compute the value of $t_*$. It is rather long—of the order of several centuries—when measured using census data[44] and a little shorter using data on tax returns[45].

These results allow us to consider very general classes of dynamics and initial conditions. We therefore conclude that in general, given a specific set of demographic conditions, the total population can grow exponentially with a common growth rate across cities. In addition, the relative population distribution at late times does not resemble anything like Zipf's law. As expected from the statements of the ergodic theorems, the population structure is set by the nature of the intercity flows (and vital rates), or equivalently by the environment **A**.

**Stochastic demographic dynamics and symmetry breaking**. The step towards a statistical solution for the city size distribution—and obtaining Zipf's law in particular—requires the additional consideration of statistical fluctuations, which must arise in the vital and migration rates. Just as in models of statistical physics, the introduction of stochasticity implies not just fluctuating quantities but potentially also the restoration of broken symmetries[46,47]. In the context of urban systems, restoring broken symmetries means removing preferences for population growth

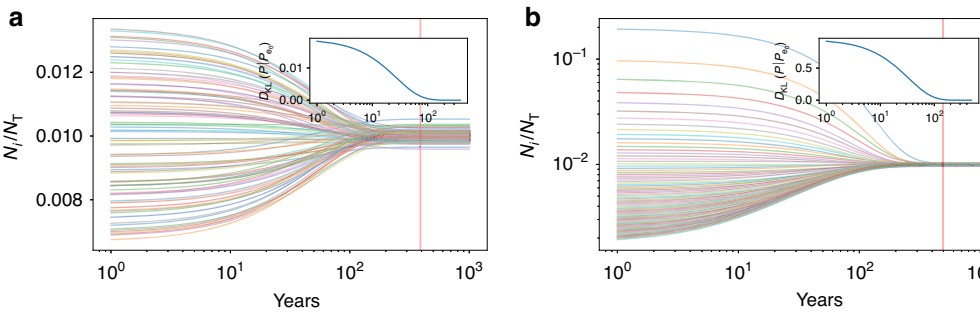

**Fig. 1 Temporal evolution of the relative population structure of cities in the same environment for different initial conditions.** Lines shows the trajectory of each city in terms of the fraction its population, $N_i$ to the total $N_T$. **a** shows an initial situation where all cities start out with similar sizes, whereas in **b** they are initiated following Zipf's law (shown in log-scale). In both cases, the relative city size distribution eventually becomes stationary at long times (vertical red line) with the same population structure. This is given by the eigenvector $\mathbf{e}_0$, corresponding to the leading eigenvalue of the environment. The insets show how the system converges in both cases to this common structure set by the environment, because the Kullback–Leibler divergence $D_{KL}(P|P_{\mathbf{e}_0})$ approaches zero at late times.

in specific cities over others, associated with imbalances in vital rates and migration flows. To make these expectations explicit we write the migration rates $m_{ij}$ as

$$J_{ij} = m_{ij}N_i(t) = \left(\frac{s_{ij} + \delta_{ij}}{2}\right)\frac{N_j(t)}{N_T(t)}N_i(t). \quad (8)$$

Eq. (8) splits the migration flows into two parts: $s_{ij}$ describes the symmetric flow between two cities ($s_{ij} = s_{ji}$), and $\delta_{ij}$ the anti-symmetric part ($\delta_{ij} = -\delta_{ji}$). Note that any correlations between $N_i$ and $N_j$ are preserved in these two quantities. Making the population size of the destination city explicit will allow to diagonalize Eq. (2) and therefore to find self-consistent solutions for the structure vector. This form of the migration currents also makes contact with the gravity law of geography, which is typically written as

$$J_{ij} = G_g \frac{N_i N_j}{d_{ij}^\gamma} = J_{ji}. \quad (9)$$

In this form, the gravity law is symmetric, corresponding to Eq. (8) when $\delta_{ij} = 0$ and $s_{ij} = G_g N_T(t)f(d_{ij})$, where $G_g$ is the 'gravitational' constant, $f(d_{ij}) = 1/d_{ij}^\gamma$ is a decaying function of distance $d_{ij}$ and $\gamma$ is the distance exponent. Only the anti-symmetric part contributes to the dynamics of population size change, since we can write Eq. (2) as

$$N_i(t+1) = \left[1 + v_i - \overline{\delta}_i\right]N_i(t). \quad (10)$$

with $\overline{\delta}_i = \sum_{j=1}^{N_c} \delta_{ij}x_j$. We can now appreciate the importance of the bi-linearity of the migration flows on both the origin and destination population as a means to diagonalize the demographic dynamics. We have achieved this however at the cost of introducing a slow non-linearity in each city's growth rate, via their dependence on the population structure vector $\mathbf{x}$. We can establish the existence of a self-consistent solution $\mathbf{x}^*$ such that for long times

$$v_i - \overline{v}^* = \sum_{j=1}^{N_c} \delta_{ij}x_j^*, \quad (11)$$

where $\overline{v}^* = \sum_{i=1}^{N_c} v_i x_i^*$, since $\sum_{i,j=1}^{N_c} \delta_{ij}x_i x_j = 0$, by the conservation of the migration currents. Note that we did not assume any explicit form for the migration flows as a function of distance between places, so that our discussion includes many specific models, including different version of the gravity law.

This equation has a clear geometric meaning: The matrix $\delta = [\delta_{ij}]$ is anti-symmetric and real, so that it behaves as an infinitesimal generator of rotations: $R(\mathbf{x}) = \mathbf{x} + \delta\mathbf{x}$. We can write the self-consistent solution in terms of these rotation as

$$R(\mathbf{x}^*) = \mathbf{x}^* + (\mathbf{v} - \overline{v}), \quad (12)$$

where $\overline{v}$ is still a function of $\mathbf{x}$ and $\mathbf{v} = [v_i]$. Note that these rotations are associated with very small angles, and that the structure vector is subject to constraints, such as staying positive. The first two terms of Eq. (12) ask for a general solution for $\mathbf{x}$ that is invariant under rotations. However, the last term, which introduces differences in relative growth rates for different cities, breaks this symmetry explicitly. This specificity of the relative growth rates leads to particular solutions that do not coincide with Zipf's law. Fig. 2a shows the example of a numerical solution in a non-linear, non-stochastic environment, defined by Eq. (8). The final population structure in this environment is still well defined, as it reaches a steady state. However, compared to the time independent case (see Fig. 1), the final structure takes much longer to unfold and has a stronger urban hierarchy.

**Symmetry restoration and the emergence of Zipf's law.** The stage is now set to consider what it takes to counter the symmetry breaking resulting from the selective structure of the environment. To do that, we must consider the statistics of the demographic rates. We start by rewriting Eq. (10) in terms of the dynamical equations for the structure vectors $\mathbf{x}$ as

$$x_i(t+1) = \left[1 + v_i - \overline{v} - \overline{\delta}_i\right]x_i(t) = (1 + \epsilon_i)x_i(t). \quad (13)$$

We now specify the relative growth rate $\epsilon_i = v_i - \overline{v} - \overline{\delta}_i$ for each $x_i$ as a statistical quantity. It is clear by construction that its average over cities vanishes, $\overline{\epsilon}_i = 0$. The only remaining question is how $\epsilon_i$ varies over time. There is strong empirical evidence that on the time scale of years to decades some cities can maintain larger growth rates than others. For example, presently in the US, most cities of the Southwest and South are growing fast, while other cities show slower growth or relative decay, such as many urban areas in the Midwest and Appalachia[48]. Moreover, it has been true in recent decades that mid-sized cities in the US have been growing faster than either the largest cities or small micropolitan areas, so that even population aggregated growth rates over certain size ranges differ. Going further back in time one could argue that the same cities of the Midwest were once booming as they became the manufacturing centers of the US, especially between the Civil War and the decades post WWII[49]. Similar arguments may be made for cities in other urban systems, such as in Europe[50,51]. Thus, it is possible that on very long time scales, $T$—on the order of a century or longer—the growth rates of cities equalize to very similar numbers and therefore the temporal average $\langle \epsilon_i \rangle = \frac{1}{T}\sum_t^T \epsilon_i(t) \to 0$. This requires that the temporal average of Eq. (11) remains zero, which means that the structure vector is both rotated and dilated 'randomly' in ways that over sufficiently long times lead to $\langle \epsilon_i \rangle = 0$, for each city. Numerical solutions of the demographic equations under these conditions show a population structure that closely resembles Zipf's law for long times, see Fig. 2b.

In the following, we show that Zipf's law is a probability density for this derived dynamics in the long run. To do this, let us characterize the variance in the temporal growth rate fluctuations as $\sigma_i^2 = \langle \epsilon_i^2 \rangle = \sigma_{v_i-v}^2 + \sigma_{\overline{\delta}_i}^2 + 2\text{COV}(v_i - v, \overline{\delta}_i) > 0$. Over the same long time scales, statistical fluctuations in the growth rate may obey a limit theorem, so that they become approximately Gaussian, with variance, $\sigma_i^2$. Then, Eq. (13) becomes simpler, and acquires a direct correspondence to familiar stochastic growth processes. It can be written in analogy to geometric Brownian motion without drift as

$$dx_i = \epsilon_i x_i = x_i \sigma_i dW \quad (14)$$

where $W$ is a standard Brownian motion. We now see how demographic dynamics can be simplified through a chain of assumptions into a form consistent with typical stochastic processes leading to Zipf's law as proposed by Gabaix[20,52].

Equation (14) can be expressed as an equation for the probability density $P \equiv P[x_i, t|x_i(0), 0]$ of observing $x_i$ at time $t$, having started with an initial condition where $x_i(0)$ was observed at time zero,

$$\frac{d}{dt}P = \frac{d^2}{dx_i^2}\sigma_i^2 x_i^2 P = \frac{dJ_{x_i}}{dx_i}. \quad (15)$$

The last term describes a probability current, $J_{x_i} = \frac{d}{dx_i}\sigma_i^2 x_i^2 P(x_i)$. This equation is exactly solvable, see Methods for technical details. To find the stationary solutions, we ask that the right hand side of Eq. (15) vanishes. The first of the two solutions corresponds to a vanishing current, $J_{x_i} = \frac{d}{dx_i}\sigma_i^2 x_i^2 P(x_i) = 0$ and is $P = \frac{c'}{\sigma_i^2 x_i^2}$, where $c'$ is a normalization constant. If $\sigma_i^2 = \sigma^2$

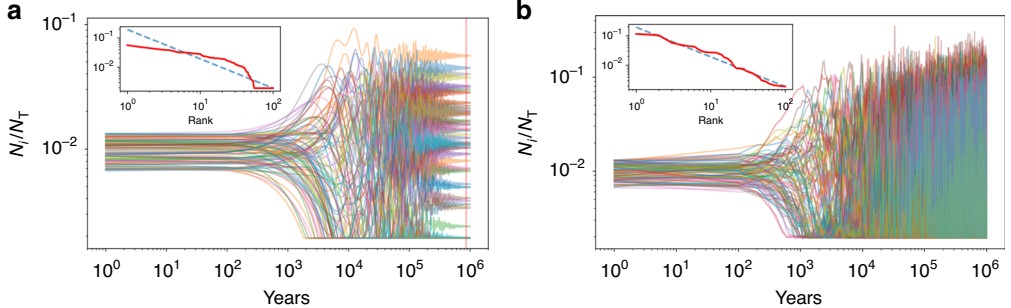

**Fig. 2 City size distribution emerging from stochastic demographic dynamics differ significantly from those in time independent environments.**
**a** depicts the temporal trajectory for a set of 100 cities in a non-stochastic, non-linear environment, described by Eq. (8). Despite generalizing the demographic dynamics in Fig. 1, this results in a fixed urban hierarchy at late times (vertical red line), regardless of initial conditions. However, when migration and vital rates become stochastic, the demographic dynamics no longer has a static solution at long times, **b**. Instead, cities constantly change their relative sizes (rank) over time. After an initial transient, the population structure fluctuates close to Zipf's law. The insets show the population structure (red) at the point in time when the non-stochastic evolution (**a**) becomes approximately stationary, with Zipf's law (dashed blue line) shown for reference.

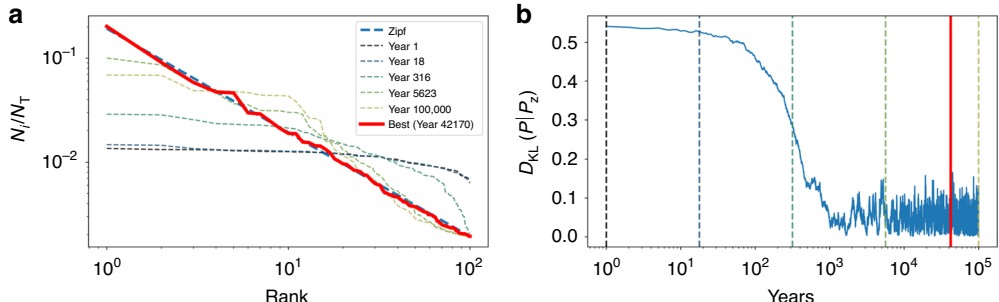

**Fig. 3 Stochastic migration rates restore the symmetry of city growth rates over time, leading to Zipf's law.** The figures show the temporal solution of the demographic dynamics (Eq. (13)) when rotational symmetry is restored through stochasticity averaged over long times. **a** Shows the population structure at different times (dashed lines) and the distribution with smallest divergence from Zipf's law over a long run (red), measured by the smallest observed $D_{KL}(P|P_z)$. **b** shows the trajectory of the Kullback-Leibler divergence away from Zipf's law over time. After a long time period, the population structure approximates Zipf's law and fluctuates around it. In this regime, at any single time only samples showing some deviations from Zipf's distribution are observable: it is only on the average over many structure vectors at long times that Zipf's law emerges as a good characterization of the city size distribution. Fig. 4 shows that this was indeed the case for the US urban system until recently.

independent of $x$, as proposed by Gibrat's law, we can drop the index so this becomes Zipf's law, $P(N) = P_z(N_i) \sim \frac{c}{N^2}$. Note for example that if $\sigma_i^2 \sim N^{-\alpha}$ ($\alpha$ may be negative), which violates Gibrat's law in terms of fluctuations of the growth rate, then $P \sim 1/N^{2-\alpha}$. Thus, the city size dependence of growth rate fluctuations will change the exponent in Zipf's law and may destroy scale-invariance altogether if it is not a power law. The second solution applies for constant current, i. e. $J_x = J$, which leads to $P = \frac{c''}{\sigma^2 x}$, where $c''$ is a normalization constant. This represents a flow of probability across the urban hierarchy. Both solutions are attractors of the stochastic dynamics that emerge for long times, $t \gg 1/\sigma^2$.

We find that in the limit where the relative difference in vital rates vanishes due to fluctuations, the structure vector becomes rotationally invariant on average and the angular symmetry of $\mathbf{x}$ is effectively restored. This leads to an effective symmetry of the relative city size distribution on the time scale $t_r = 1/2\sigma^2$, when growth rate fluctuations vanish. The time $t_r$ can be estimated using US data to be typically very long, on the range of many centuries or even millennia. Under these conditions all cities share statistically identical dynamics and can be interchanged, resulting in Zipf's law as the time average of population size distributions, see Fig. 3. In the same limit, the anti-symmetric components of intercity flows will average out and the gravity law

will emerge in its conventional form. However, the observations of city sizes at each time are samples of this distribution and may reflect decade-long observable positive and negative preferences for certain cities. Figure 4 shows this effect for the US urban system using decennial census data from 1790 to 1990. For time periods of a few decades, the temporal average does not visibly converge to Zipf's law: The blue line in the inset of Fig. 4 depicts this effect. However, the cumulative temporal average of the size distributions starts to approximate Zipf's law after about five decades. The red line (inset) shows the KL divergence between the cumulative temporal average and Zipf's law, starting from 1790 to subsequent census.

## Discussion
We have shown that the demographic dynamics of urban systems can result in different city size distribution and how Zipf's law, the gravity law and other coarse-grained statistical regularities in geography emerge approximately from these dynamics as averages when certain symmetries are restored over sufficiently long times.

Several of the properties of Zipf's law that have been controversial in the literature can be clarified from this perspective. First, Zipf's law is not a "unique signal for the integration of cities in an urban system"[54]; it is merely one distribution of city sizes

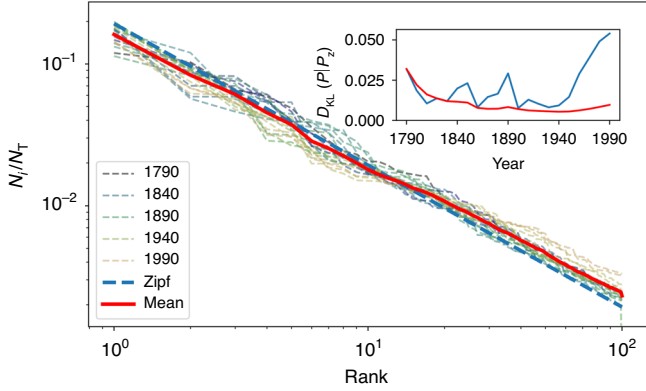

**Fig. 4 The dynamics of the rank-size distribution for cities in the US between 1790 and 1990.** Population structure distributions for the largest 100 cities in the US[53] are shown as dashed lines, while the average over all years is shown in red. The inset depicts the $D_{KL}(P|P_z)$ to Zipf's law of the population structure for each available year (blue) and of the cumulative average over all structure vectors up the specific year (red). We see that the temporal average steadily approaches Zipf's law as more years are added. After an initial phase, the average is always closer to Zipf's law than the distribution in any single year. Note that in recent decades, the population structure vector began consistently deviating from Zipf's law, leading also to a small divergence of the cumulative temporal averages. This effect is to a large extent the consequence of mid-sized and large metropolitan areas growing faster over this period than the largest cities in the US urban system.

among many less symmetric ones that result from such integration. We have shown in this respect that integration of an urban system depends on strong intercity migration flows—but not necessarily the symmetric ones that might lead to Zipf's law. Intuitively, such flows mix people up between places creating a single common dynamics across all cities that is not associated with any particular kind of economic or political organization beyond enabling free intercity migration. However, asymmetries in the migration flows (and, thus, deviations from Zipf's law) might originate in systemic properties, but can have a manifold of other reasons—such as socio-economic opportunities, natural disasters, or demographic change—or originate from combinations of them.

Second, in a recent article Cristelli et al.[54] make the important point that the properties of Zipf's law are systemic and cannot be understood simply on the basis of its mathematical form as a power law. They show that Zipf's distribution is coherent in that subsamples of a population distributed according to Zipf's law do not reflect the same statistics. They also emphasize that because the largest city in the system sets the overall properties of rank, the distribution makes no sense without it, something they call screening. These properties can be understood through the conservation of the current $J_x$ and its dynamical consequences. If we replace any other probability distribution, including a normalized subsample of Zipf's law, into the current we will obtain a non-zero drive to the entire system dynamics because $P[x]$ is a field in $x$. This systemic response signals a lack of coherence and eventually restores $P[x]$ back to Zipf's distribution. Similarly, a set of cities organized such that their relative populations has $J_x = 0$ ensures that other cities cannot easily take their sizes and ranks, leading to screening as a result of the global conservation law of the probability current. Under these conditions there can only be churn in the relative positions of cities in the rank hierarchy. The characteristic extinction time for a city with initial rank $r$ to fall below the lower boundary $x < x_m$ can be read from the

solutions in Methods to be $\langle \Delta t(r) \rangle \sim \frac{1}{2\sigma^2} \ln \frac{r_{max}}{r} = t_r \ln \frac{r_{max}}{r}$. The longest time, for the largest city to fall off the system, is $\langle \Delta t(r = 1) \rangle t_r \ln N_c$, which is the product of the characteristic time for the reversal of city growth rates, $t_r$, multiplied by $\ln N_c = \ln r_{max}$, which is a measure of the depth of the urban hierarchy.

To better appreciate the commonly observed deviations of the largest and smallest cities in the distribution, consider that the current can be integrated over any finite range of $x \in [x_l, x_u]$ to give

$$\int_{x_l}^{x_u} dx J_x = x_u^2 P[x_u] - x_l^2 P[x_l] = 0 \rightarrow P[x_u] = \frac{x_l^2}{x_u^2} P[x_l], \quad (16)$$

which is clearly satisfied by Zipf's distribution with $z = 1$. This shows how the statistical dynamics of population sizes depends on values at the extremes of **x**, the maximum, $x_M$ and minimum $x_m$. While it may be natural to set $x_M = 1$ with a small probability, it is more problematic to choose $x_m$'s value, thereby defining the population of the smallest possible city in the sample. In particular, because $x_m = N_m/N_T > 0$, we must specify how places with population $N < N_m$ relate to our distribution at the lower boundary. The vanishing current condition tells us that there should be a Zipfian "ghost" distribution of cities, such that $P[x < x_m] = \frac{x_m}{x} P[x_m]$, with lots of small cities.

Expanding on a point stressed by Saichev et al.[5], this means that the boundary conditions, especially for small cities, are critical in shaping the resulting distribution. This issue betrays a failure of true scale-invariance hidden in Zipf's law. The solution of Eq. (14) in the absence of the vanishing current is a lognormal distribution[5] with average given by $\ln x(t) = \ln x(0) - (\sigma^2/2)t$ (see Methods). This means that, in the absence of boundary conditions at $x_m$, the distribution will become more and more peaked at smaller and smaller values of $x$ over time. The conservation of the current stops this decay from happening, because it effectively injects some compensatory probability for small cities getting smaller, from other even smaller cities (not in the sample) getting larger. This logic requires an immense number of smaller and smaller cities. For example, if like in the US or China the largest city in the system has ~20 million people, one needs 20 million cities with one person, 2 million with 10, 200,000 with 100 and so on, which is not observed. This inevitable deficit of small towns relative to Zipf's law will always lead to a loss of scale invariance for high ranks and the need to understand the nature of demographic processes at play in this regime on the basis of more fundamental processes, returning us back to the demographic dynamics of Eq. (2).

Zipf's law acquires a special status among other possible distributions in light of the property of neutrality, which emerges as the rotational symmetry of the dynamics is restored over long times. The term refers to the absence of selection in population dynamics, see Methods. As such, resulting distributions are maximally disordered given additional constraints (maximum entropy). This means that deviations from Zipf's law express selection of people towards certain places (resulting from temporary $\epsilon_i \neq 0$) and should be measured in terms of information. In this respect, we can measure the surprise of an observed state of the system relative to the Zipfian expectation as $S(x) = \log \frac{P[x]}{P_z[x]}$, and the total information (total surprise) in the choices underlying the actual observed distribution relative to the neutral situation (Zipf's law) in terms of the Kullback-Leibler divergence $D_{KL}(P|P_z) = \overline{S}$, as we have done in Figs. 1, 3, and 4. Those measures additionally provide a way to compare structures of systems at different times and can reveal temporal trends, signaling basic structural changes in the underlying demographic processes. Fig. 4, for example, depicts a strong and long lasting

increasing deviation from Zipf's law for the US. It might be a signal that the US is in a transitory state with strong preferences for certain regions since the 1940s, such as mid-sized cities in Texas and the Southwest. However, such statements for specific urban systems need further context-specific investigation, which is beyond the scope of this article.

In summary, starting with the most general demographic equations, we can derive many instances of integrated urban systems with city size distributions that differ substantially from Zipf's law. We can understand the process by which universal patterns in geography emerge as a sequence of situations that restore the symmetry of the demographic dynamics and ultimately rely on averaging stochastic behavior of vital and migration rates over sufficiently long times. Seeing Zipf's law and other laws of geography in this light helps us appreciate the information content of associated deviations in real urban systems, and trace them back to specific choices and preferences associated with people's agency in terms of births, deaths, and migration. Formally, the approach developed here for urban systems relies only on the analysis of the transition probability between types against a general background of multiplicative growth dynamics. As such, it is very general and readily applies to other situations in complex systems where rank-size rules are also observed approximately.

## Methods

### Probability solution for geometric random growth with boundary conditions.
The Fokker-Planck equation for random geometric growth without drift is

$$\frac{dP}{dt} = \frac{d^2}{dx^2}\frac{\sigma^2}{2}x^2 P, \tag{17}$$

where $P = P[x(t), t | x(0), t_0]$ is the conditional probability of observing state $x$ of the random variable at time $t$, given the initial state $x(t_0)$ at time $t_0$. For simplicity of notation, we have dropped the $i$ indices in $x$ and $\sigma$ and write $x(t_0)$ as $x_0$.

Equation (17) has some similarities with the diffusion equation in physics and can analogously be solved exactly. First, it is useful to change variables so as to eliminate the non-linear term $x^2$. We set $y = \ln\frac{x}{x_0}$ and $\tau = \frac{\sigma^2}{2}(t - t_0)$. By changing the variables in Eq. (17) we find

$$\frac{dP}{d\tau} = \frac{d^2P}{dy^2} + 3\frac{dP}{dy} + P \equiv P'' + 3P' + P. \tag{18}$$

This equation is now linear and can be solved in two ways. The first way is using factorization and solve it as a heat equation

$$P = e^{-\frac{3}{2}y - \frac{1}{4}\tau}g(y, \tau) \tag{19}$$

and

$$\frac{dg}{d\tau} = g''. \tag{20}$$

The second way is to solve it directly via a Fourier transform, so that

$$P[y, \tau] = \int dk\, e^{iky}P[k, \tau], \tag{21}$$

which leads to

$$\frac{dP[k, \tau]}{d\tau} = (-k^2 + 3ik + 2)P[k, \tau]. \tag{22}$$

This equation can be solved via a separation of variables, $P[k, \tau] = f(k)T(\tau)$, which leads to an eigenvalue problem:

$$\frac{dT}{d\tau} = -wT, \qquad (-k^2 + 3ik + 2)f(k) = -wf(k). \tag{23}$$

Solving for $k$ we obtain:

$$k = \frac{3}{2}i \pm \frac{1}{2}i\sqrt{1 + 4w}, \tag{24}$$

with

$$P[y, \tau] = \int dk P[k, 0]e^{iky - (k^2 - 3ik - 2)\tau}, \tag{25}$$

where $P[y, 0] = f(k)T(0)$. In particular, there are two stationary solutions, for $w = 0$, with $k = k_0 = 2i$ and $k = k_1 = i$. Substituting $k_0$, we see that the solution corresponds to $P \sim e^{-2y} = \frac{1}{x^2}$, which is Zipf's distribution. The other solution

corresponds to the existence of a constant probability current up or down the urban hierarchy associated with different boundary conditions, see main text.

In order to obtain Zipf's law as the stationary distribution for long times, we must add constraints to the geometric random growth dynamics. To see this more explicitly, we consider the full dynamical solution, which can be written in terms of $y$ as,

$$P[y, \tau] = \alpha_1 e^{-2y} + \alpha_2 e^{-y} + \int dk a_k e^{iky - (k^2 - 3ik - 2)\tau} \tag{26}$$

To obtain Zipf over the long run, three conditions are needed: First, the integration constant $\alpha_2$ has to be zero when the probability current is set to zero. In terms of $y$, this reads as

$$J(y) = e^{-2y}\frac{d}{dy}\sigma^2 e^{2y}P[y, \tau] = 0. \tag{27}$$

Second, the constant $\alpha_1$ can now be fixed by normalization of Zipf's distribution as a probability:

$$\alpha_1 \int_{x_m}^{x_M} dx\frac{1}{x^2} = 1, \tag{28}$$

thus $\alpha_1 = \frac{x_M x_m}{x_M - x_m}$, where $x_M$ and $x_m$ are the upper and lower boundaries of city sizes, respectively. A natural choice for the upper boundary is $x_M = 1$, whereas the choice for the lower boundary is not obvious, see main text. Third, we can now set the boundary condition on the time varying solutions, which are now

$$g(y, \tau) = P[y, \tau] - \alpha_1 e^{-2y} \tag{29}$$

and must be taken to vanish at the boundaries in $y$. The integral over the range of city sizes needs to be zero to preserve the probability normalization. For example, by asking that these solutions are real and vanish at the boundaries, meaning $g(y = y_m, \tau) = g(y = y_M, \tau) = 0$, we get

$$g[y, \tau] = \sum_n a_n \sqrt{\frac{2}{y_M - y_m}}\sin k_n[(y - y_m) - 3\tau]e^{-(k_n^2 - 2)\tau}, \tag{30}$$

with $k_n = \frac{2\pi}{y_M - y_m}n$ and $n = 1, 2, 3, \dots$. The coefficients $a_n$ are determined via the initial condition $g[y, 0]$ in the usual way. The sine functions are an orthonormal basis under integration over the domain of $y$, thus

$$a_n = \sqrt{\frac{2}{y_M - y_m}}\int_{y_m}^{y_M} dy g[y, 0]\sin k_n(y - y_m). \tag{31}$$

Even though these functions now obey the boundary conditions, the temporal structure is similar to the case with no boundary conditions and the decay of the initial amplitudes occurs on a time scale set by $\tau$, which is $t - t_0 = \frac{1}{2\sigma^2}$ and can be very long for small volatilities, $\sigma$.

### Lognormal solution in the absence of conserved current.
We can find the general solution in Eq. (25) with the initial condition $P[y, 0] = \delta(y)$, which corresponds to the case $N(t = 0) = N_T$. Then $P[k, 0] = 1$ and the solution for all times is

$$P[y, \tau] = \int dk e^{iky - (k^2 - 3ik - 2)\tau} = \frac{e^{-y}}{\sqrt{2\pi\tau}}e^{-\frac{(y + \tau)^2}{4\tau}}. \tag{32}$$

Returning to our original variables this becomes

$$P[x, t | x_0, t_0] = \frac{x_0}{x}\frac{1}{\sqrt{\pi\sigma^2(t - t_0)}}e^{-\frac{\left(\ln\frac{x}{x_0} + \frac{\sigma^2}{2}(t - t_0)\right)^2}{2\sigma^2(t - t_0)}}, \tag{33}$$

which is a lognormal with log-mean $\langle \ln x \rangle = \ln x_0 - \frac{\sigma^2}{2}(t - t_0)$, and log-variance $\langle (\ln x - \langle \ln x \rangle)^2 \rangle = \sigma^2(t - t_0)$. As a result for late times, in the absence of a boundary condition that sets the current $J_x$, the city size distribution becomes peaked around smaller and smaller sizes, and become broader and broader.

We can define an "extinction time" $\langle \Delta t(r) \rangle$ as the expected time interval for a city of initial rank $k$ to fall through the lower boundary condition, at $x = x_m$. This is defined implicitly through the conditional probability $P[x_m, t; x_0, t_0] = p \sim 1$, where $x_m$ is the minimum size, which close to Zipf corresponds to maximal rank $r_{max} \sim 1/x_m$. Observation of the exact solutions above tells us that the leading time is $\langle \Delta t(r) \rangle \sim \frac{1}{2\sigma^2}\ln\frac{x}{x_m} = \frac{1}{2\sigma^2}\ln\frac{r_{max}}{r}$. There are subleading terms that depend on the exact initial condition. The longest extinction time belongs naturally to the largest city (with initial $r = 1$): it is $\langle \Delta t(r = 1) \rangle \sim \frac{1}{2\sigma^2}\ln r_{max}$, which depends on how many cities there are in the urban system because $r_{max} = N_c$.

### Neutrality of population dynamics and Zipf's law.
In evolutionary population dynamics, a change in the relative probability of types (in the absence of errors) is attributed to selection. The situation when selection is absent and only statistical fluctuations drive the dynamics is known as neutral dynamics.

We noted in the main text that the structure vector $x_i(t) = \frac{N_i(t)}{N_T(t)}$, $i = 1, \dots, N_c$ is the probability of finding a person in city $i$ at time $t$ out of a total population

$N_T(t)$, across all cities. In evolutionary population dynamics we write the evolution of this probability as

$$x_i(t + 1) = w_i(t)x_i(t) \qquad (34)$$

The quantity $w_i(t)$ is the fitness of state $x_i$, because if $w_i(t) > 1$ this state becomes more common in the updated population and vice versa when $w_i(t) < 1$. Note that by normalization of the updated frequency the population average $\sum_i x_i(t + 1) = 1 = \sum_i w_i(t)x_i(t) = \bar{w}(t)$. Comparing to Eq. (13), this establishes that $w_i = 1 + \epsilon_i$. Positive (negative) selection correspond to the situation when $w_i > 1$ ($w_i < 1$). Neutrality corresponds to $w_i = 1 \rightarrow \epsilon_i = 0$, which we showed to be necessary to obtain Zipf's distribution. Consequently, Zipf's law is a neutral distribution.

Note that Zipf's law is not the only neutral distribution for multiplicative dynamics, but becomes unique when we impose the additional condition, $J_x = 0$.

Additionally $\ln w_i$ has a meaning as information. To see this write,

$$\log w_i \simeq \epsilon_i = \log \frac{x_i(t + 1)}{x_i(t)} = \log \frac{x_i(t + 1)P(\mathbf{A})}{x_i(t)P(\mathbf{A})} = \log \frac{P(x_i, \mathbf{A})}{x_i P(\mathbf{A})}, \qquad (35)$$

where $P(\mathbf{A})$ is the probability of a particular environment and $p(x_i, \mathbf{A})$ is the joint probability of the population structure and of states of the environment. The idea is that the updated probability is the structure vector given (conditional on) the influence of the environment, as we wrote in Eq. (13), so that $x_i(t + 1) = P(x_i|\mathbf{A}(t))$. Thus, $\log w_i$ once averaged over $x$ and $\mathbf{A}$ is the mutual information between the population structure and the environment. When the evolution is neutral $P(x_i, \mathbf{A}) = x_i P(\mathbf{A})$, the two variables are statistically independent and there is no information from the environment being encoded in the population structure. Since the environment, in our case, is the space of preferences in vital and migration rates, there is no structure of these preferences encoded in the population structure when we observe Zipf's law. Only the deviations from it, when $\epsilon_i \neq 0$, can in this sense give us information.

**Time integration**. Numerical solutions shown in Figs. 1–3 where obtained by iterating the respective environments, $\mathbf{A}$, starting with several different initial population states, $\mathbf{N}(0)$, as expressed by Eqs. (3) and (5). We assumed that all environments are strongly connected graphs; the migration probability from city $i$ to city $j$ is non-zero, or $A_{ij} > 0$. These parameterizations reflect features of real world urban systems such as the average annual fraction of population that moves between metropolitan areas in the US in the last few decades, which is about 1.8%, according to US Census Bureau[44] and tax returns reported by the Internal Revenue Service[45]. The data from both sources shows that the migration rates $\delta_{ij}$ follow a lognormal distribution. We implemented the stochastic environments to reflect this statistical distribution.

The introduction of stochasticity in Figs. 2 and 3 requires that we set boundary conditions at large and small $x$ as discussed in the main text. The upper bound is less critical and a natural choice is to allow the whole population to concentrate in one city, or $x_M = 1$. This state has very low entropy, so that it is extremely rare for it to occur by chance. In practice, it is never observed in numerical solutions even when stochastic fluctuations are strong. On the other hand, the boundary condition for small cities is violated all the time, as shown in Fig. 2. Varying the value of $x_m$ shows that the best value for the lower bound is of the order of the size of the smallest city according to Zipf's law ($N(r = N_c) = N_0/N_c$), resulting in:

$$x_m = \frac{N_0}{N_c N_T}. \qquad (36)$$

When a city's time evolution violates this lower constraint, our implementation resets the population size of the city back to its former size, typically just above $x_m$, and adapts the other cities so that

$$\sum_{i=1}^{N_c} x_i = 1 \qquad (37)$$

This is achieved by reducing the $x_i$ uniformly among all other cities without violating the lower bound during this process. This implementation of the boundary condition for small cities is necessary to obtain Zipf's law. It mimics the probabilistic effects of a ghost population of very small cities, as described in the main text.

**Data**. Data was obtained from two main sources, the US Census Bureau (USCB) and the US Internal Revenue Service (IRS). USCB provides a data set for the complete population in the US on a decennial basis. We use CPS Historical Migration/Geographic Mobility Tables[44] as the main empirical basis in this paper. IRS data is released annually[45]. Migration rates are only implicitly given in this dataset. It provides locations of tax filings every year and corresponding (different) locations the year before. IRS data does not capture the whole population, since only the portion files tax returns (perhaps 85%). It nevertheless provides valuable insights into the migration patterns within the US.

In both cases data are provided at the county level. We aggregate the data to the level of Metropolitan Areas (MSA). MSAs are functional cities, defined as clusters of contiguous counties that have strong socio-economic ties to an urban core, measured via commuting fluxes. For this reason, MSAs are definition of urban

areas as integrated labor markets. A crosswalk from counties to MSAs is provided by the National Bureau of Economic Research (NBER)[55].

This data allows us to build all entries of the environmental matrices $\mathbf{A}$ and their migration flows. The net vital rates are given implicitly in both data sets via the total population change for each county in every year. We assume that the parts of the growth rates that can not be explained by national migration, stem from the balance of births, deaths, and international migration. All references to data are based on the matrices constructed this way on the basis of USCB and IRS data, except for Fig. 4, which is based on another data set from USCB[53].

## Data availability

The data that support the findings in this study are available from the following sources:

The main empirical basis on migration and geographic mobility data are available from USCB: https://www.census.gov/data/tables/time-series/demo/geographic-mobility/historic.html.

Migration data from IRS are available at https://www.irs.gov/statistics/soi-tax-stats-migration-data.

The county-MSA crosswalk are provided by NBER and available at https://data.nber.org/data/cbsa-msa-fips-ssa-county-crosswalk.html.

The source data underlying Fig. 4 are available are https://www.census.gov/population/www/documentation/twps0027/twps0027.html.

## Code availability

Code to create all figures is available at https://github.com/mansueto-institute/DemSymNEmergenceOfUniversalPatterns.

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

## Author contributions

L.B. and D.Z. conducted the research, L.B. solved the Fokker-Plank equation, D.Z. implemented the numerical solvers, L.B. and D.Z. wrote and edited the paper.

## Competing interests

The authors declare no competing interests.
