## [Peer Review File · Nature Communications]

Reviewers' comments:

Reviewer #1 (Remarks to the Author):

In this article, the authors propose to show how the statistical form of the population size distribution of cities can be deduced from the modalities of change of the two demographic components of growth, namely movements of births and deaths and migration flows.

But one discovers from reading the article that the intention of the authors is not to deepen the real demographic processes that make up urban growth, but rather to engage in an exercise full of mathematical virtuosity (I do not have the skills required to judge its accuracy) to derive a form of distribution of city sizes, at the cost of simplifying assumptions that add nothing to our knowledge of the cities, and which also do not allow to hope to bring closer the proposed model of empirical observations made on these processes. Thus, they refer to an old ergodic theorem used in demographic theory to compute stationary populations, grouping under the term "environment" all that could constitute a rich description and adapted to the urban questions of the behavior of the populations (for example the degree of mobility, and more importantly the spatial configuration in which inter-urban migrations take place). The choice of the term "environment" seems to me excessively questionable because it can lead readers interested in the cities toward other interpretations.

In their model presented page 4, the authors without any discussion nor empirical evidence make an important assumption "international migration to city i , as well as to and from non-urban places, can be incorporated into the vital rates". Knowing the important role of migrations, either international in the case of US, or from rural areas towards cities at some historical periods of their development in all countries of the world, this can hardly be accepted.

The trick is obvious in this passage on page 9 where the register of communication which passed smoothly equations of quantum physics with breaking symmetry to the ranking of size of the cities may perhaps appeal to some followers of physics theoretical but does not spare the understanding of the specialists of the cities (if they had the patience to read so far):

« There are some analogies to Eq. 6 from other systems that may provide us with some additional intuition. The dynamics of Eq. 6 is analogous to a Schrödinger Equation in imaginary time, with Hamiltonian $H = A \square I$, where I is the identity matrix. Such dynamics are dissipative, converging to a zero temperature solution determined by the ground state of the interaction Hamiltonian H . The dominant eigenvector of this matrix plays the role of this ground state to which the dynamics converges to at long times. This "ground state" breaks the symmetries of the initial state and determines in principle any specific city size rank ordering resulting from the interaction structure between cities, m_{ij} »

Another example on page 14 when the introduction of stochasticity in the growth model is commented in these terms:

« This represents a flow of probability across the urban hierarchy and is somewhat analogous to the solution of the heat equation on a conducting one-dimensional bar with boundary conditions at different temperatures, implying a flow of energy through the medium ».

One can understand and approve the deliberate undertaking of authors to found a physics of urban systems that is universal (in the sense that it takes as little as possible account of historical and social evolution) through the transfer of established models for the physical sciences. But in fact this is a veneer and not a real transfer, because when the authors give some examples "urban" (eg page 11), nothing allows the reader to discover how the banal statement that it represents (cities at times grow faster than others) is illuminated, or transformed, by all the mathematical discourse that precedes. The call to urban references is here cosmetic, it serves to illustrate a mathematical reasoning, which is not really confronted with a general state of knowledge about the cities or the demographic processes of growth of their populations.

I think the concluding sentence that best reveals the contradictory nature of this posture is:
« Seeing Zipf's law and other laws of geography in this light helps us appreciate the information content of associated deviations in real urban systems, and trace them back to specific choices and preferences associated with births, deaths, and migration. The approach developed here for urban systems relies only on the analysis of the transition probability between types against a general background of multiplicative growth”.

Which of course brings nothing more to the geographer who deals with these questions. That is sad, because geographers for instance have demonstrated in other studies the role of a series of factors explaining major variations in the shapes of the observed distributions of city sizes that are much less vague than the invocation of “specific choices and preferences associated with births, deaths, and migration”.

Details :

The idea that Zipf's law always has an exponent 1 should be removed from the abstract and the line following equation 1, Zipf himself had foreseen variations of the exponent and the literature abounds with concrete examples of these variations, explanations have even been proposed.

Page 2 the review paper by Clémentine Cottineau could be quoted : Cottineau, C. (2017). MetaZipf. A dynamic meta-analysis of city size distributions. PloS one, 12(8), e0183919.

With regard to the adjustments of the city size distributions in Europe (page 3 ref 38), would it be possible to quote European rather than American references?

Error in the bibliography on the name of François Moriconi-Ebrard

In summary I am not convinced of the interest of this article, whose content does not match its title. It does not produce “universal patterns for urban systems”, at most it proposes another way to generate various forms of Zipf's law without really being able to link them to socio-historical processes of population growth through natural and migratory balances. A little more modesty in the title would be required. I leave it to the mathematical evaluators to say if the article brings something more to the immense literature on this form of distribution, whereas for the specialists of the cities the contribution does not seem indispensable. In any case, the authors do not really seem eager to engage in dialogue with them.

Reviewer #2 (Remarks to the Author):

In the paper "Demography, Symmetry and the Emergence of Universal Patterns in Urban Systems" by Bettencourt and Zuend, the authors use population dynamics to derive the distribution of city sizes. They consider a range of models of increasing complexity (from linear deterministic to nonlinear deterministic to nonlinear stochastic) and show that the latter may lead to a Zipf-like distribution. The authors address the question of deviations from Zipf's laws.

I think this is a very interesting paper that studies an interesting question by methods that come from statistical physics and applied math. I have several questions for the authors.

(1) In the model considered, the number of cities remains the constant, while the authors talk about very long time-scales. During these time-scales cities die and and new cities appear. How can this be incorporated? Will it change the analysis and conclusions?

(2) To derive the Zipf's law, the authors assume that the rate of change, epsilon, in equation (12), has the time-average going to zero. How can such an assumption be justified? The authors talk about some cities growing faster than others during certain periods in history, and then slower at later or

earlier times, but to me this doesn't sound like a good argument for this mathematical assumption. Further, ϵ depends on x , which is a solution of the equation. Can we even make such an assumption logically? (I suppose one could assume this, solve the equation, and then show that the assumption was true, but I don't think this has been done here, please clarify if I am wrong). If the fact that the mean of $\epsilon \rightarrow 0$ as $t \rightarrow \infty$ is an assumption of the model, then what happens to the results when this assumption is relaxed? (It appears to be a crucial assumption for the analysis).

(3) The population of people has been growing exponentially on the time scales of interest. How is this compatible with the assumption of the growth rates averaging out to zero?

(4) The authors mention using data at several points in the paper (e.g. to justify their assumptions, to estimate the various scales, etc) -- but it is not really explained how the data were used. It would be important to make these procedures more transparent.

(5) The authors talk about numerous deviations from Zipf's law that have been reported. On page 16 they state: "It is only on the average over many structure vectors at long times that Zipf's law emerges as a good characterization of the city size distribution." This appears to be the central result of the paper: even though Zipf's law doesn't really hold at any particular moment of time, it holds on average. In order to relate the theoretical findings to the subject of the paper, it would be necessary to find evidence of this in real data. The authors use data to validate parts of their model -- can these same data (at different time points) be used to illustrate that the Zipf's law really holds if we average things out in time?

Apart from these major comments, there is a number of additional questions, listed in the order of their appearance in the text:

Page 7 last line: there shouldn't be an arrow inside the parentheses.

Fig 1: Why do all the cities converge to very similar sizes here?

The material on page 8 is standard and may be moved to an appendix or methods.

Page 9: reference to the Schroedinger equation is unnecessary, as one is familiar with this type of behavior from a wide variety of other areas (e.g. linear/linearized ODEs and PDEs, pattern formation, turbulence, etc). I am referring to eventually converging to the leading eigenvector with the speed determined by the 2nd eigenvalue.

Page 9: "the introduction of stochasticity implies not just fluctuating quantities but potentially also the restoration of broken symmetries" -- this statement is not clear to me.

Page 11, top lines: if v is a vector whose components are v_i , how is it a function of x ?

Same page: "The first two terms ask for a general solution for x that is invariant under rotations." -- the first two terms of what? Equation (10)? Also, what does this type of symmetry mean if we think of cities and their populations? Is there a clear intuitive meaning of this symmetry and its breaking?

"However, compared to the time independent case, the final structure takes much longer to unfold." -- do you mean, compared to the linear case?

Also, you talk about a self-consistent solution earlier, do you find it explicitly here, or are only numerical solutions available? Is convergence to a unique steady state independent of the initial

conditions proven in the nonlinear case? What defines the speed of convergence? (I realize these are hard questions in a nonlinear system, I just want a clarification of what has actually been done here).

Page 11, eq (12) -- where does the additional term, $-\bar{v}$, come from (compared to eq (10))? I understand that the authors aim to introduce stochasticity in the coefficients, so is now v_i stochastic? Or possibly deltas?

Fig 2: The non-stochastic system produces the distribution which is sort of similar to the Zipf's law. Is this a coincidence? Can it be very different under different assumptions on the coefficients?

Page 15, 1st paragraph: please provide details of how the time-scale was estimated from data.

Page 17: end of 1st paragraph: I thought this estimate of convergence was only applicable to linear systems?

Discussion: more technical bits should be reserved for other sections, here it would be nice to explain the finding in intuitive way and relate them with observations/data.

Reviewer #3 (Remarks to the Author):

The paper shows how the structure of migration flows between cities together with the differential magnitude of their vital rates determine a variety of size distributions. These results provide a framework for deriving various size distributions under specific conditions and help resolve problems associated with the deviations of these distributions in terms of symmetry, information, and selection. The authors provide a series of demographic processes that may result in different city size distributions and explain in what way each process affects the resulted distribution. To that end, the paper is innovative and thorough, and it sheds light on the fact that despite the great attention Zipf's law has been given, it is one option of city size distribution among many others. The work is presented in a convincing and explanatory way and the paper is well-read. The topic of size distributions is of interest to a wide community of scholars from various disciplines.

Having said that, I have a few questions that the authors may want to address (while I do believe these questions can be explained by the proposed model, they are not addressed in the text):

1. The authors mention the possibly long periods of time it may take the system of cities to reach their final structure. As for long periods of time, the system of cities themselves may change (due to changes in the boundaries of the country for example) – how does the model addresses such changes? This question may also be relevant to the disappearing of cities due to wars or natural disasters. This issue is different from inter-national immigration as large cities can disappear or appear in the system at a unique point in time.

2. In the last decades, the discourse on urban systems has crossed the national boundaries, and currently, there is a system of world-cities that is independent (in many ways) on the national association of each city. In what way does the proposed work account for the growth of these cities (that are typically the largest cities that also govern the overall properties of the rank, as mentioned in the paper).

3. Does the issue of scale (i.e. the physical size of the country) is also considered in terms of intervening opportunities (in addition to the gravitation constant)? See:

Simini F, Gonzalez MC, Maritan A and Barabasi A-L (2012) A universal model for mobility and migration patterns. Nature 484: 96–100. doi: 10.1038/nature10856 PMID: 22367540

I also think the paper may benefit from addressing the following work:

1. A. Blank, S. Solomon, Physica A: Statistical Mechanics and its Applications 287 (2000) 279.

2. L. Benguigui, M. Marinov. arXiv preprint arXiv:1507.03408 (2015).
3. L. Benguigui, M. Marinov. arXiv preprint arXiv:1607.00856 (2016).
4. L. Benguigui, E. Blumenfeld-Lieberthal. Physica A: Statistical Mechanics and its Applications 384.2 (2007): 613-627.

Thus, I recommend to accept the paper with minor revision

Reviewer 3

The authors mention the possibly long periods of time it may take the system of cities to reach their final structure. As for long periods of time, the system of cities themselves may change (due to changes in the boundaries of the country for example) – how does the model addresses such changes? This question may also be relevant to the disappearing of cities due to wars or natural disasters. This issue is different from inter-national immigration as large cities can disappear or appear in the system at a unique point in time.

This is a question we should have clarified in the original manuscript. Thank you.

If the total number of cities changes, the matrix **A** will need to change size adding or subtracting lines and columns. The dynamics of the system may include latent cities like this, if their population sizes are initially zero, or by setting their entries in matrix **A** to zero initially or after they disappear. No essential element of the analysis changes. While we do not show this in the numerical solutions, it is straightforward to adapt the environment at any given point in time to add or remove cities from the system.

Cities in the urban system at some point in time, are also allowed to disappear (and be replaced by others) if necessary. If sudden, this can be modeled as a large fluctuation in the population dynamics, for example with an extreme negative growth rate for the specific place including large out-migration (e.g. New Orleans during Katrina). Given enough time, Fig. 2b shows that some cities experience such dramatic growth rates fluctuations at certain points in time (Oil rushes are also a phenomenon of this kind in small US cities, for example in Alaska). The model, being demography, naturally describes such extreme events: They might actually support the restoration of symmetry over time.

In cases of disasters, cities normally do not disappear completely (again think of New Orleans). They might nevertheless decrease in status in the urban hierarchy due to loss of population, but will still exist in the ensemble. This, in fact, is how the lower boundary condition can intuitively be understood. The lower boundary condition allows for the shrinking, disappearing, and (re)appearing of new cities.

As mentioned in the main text and the methods section, the conservation of this current is necessary to obtain Zipf's law, as it injects small compensatory probability for small cities becoming smaller and small cities – from outside the ensemble – becoming bigger. There are no restrictions on how fast this process might happen. Additionally, there are no specific restrictions on the number of cities in the system.

In the last decades, the discourse on urban systems has crossed the national boundaries, and currently, there is a system of world-cities that is independent (in many ways) on the national association of each city. In what way does the proposed work account for the growth of these cities (that are typically the largest cities that also govern the overall properties of the rank, as mentioned in the paper).

This is an excellent question. Thank you.

The question to which "system" a city belongs to is clarified by the demographic dynamics we introduce in the paper. Note that the approach is not focused on a specific type of urban system and may includes cities across national boundaries. We just start

out with a set of cities and their reciprocal migration flows.

The answer is that although candidate "world-cities" in the US – such as New York City and Los Angeles – certainly receive a substantial number of international migrants, their total migrant flows are vastly dominated by internal migration to/from other US cities. Because it is these reciprocal exchanges that determine – as we show in the paper – if a set of cities behaves like an integrated urban system, we can therefore conclude that although connected to some extent, it is probably wrong to include international cities and US candidate "world cities" as part of the same urban system. Moreover, two cities are not part of the same urban system if migration is predominantly in one direction; it takes strong reciprocal links to create such integration. The formalism of the paper allows us to test these questions quantitatively, but we are not aware of available data that allows us such tests at this point. We introduced a number of comments in the paper about these issues.

There are a few additional – more indirect – arguments that we can make with current evidence that point to the lack of integration of US large urban areas with other cities internationally. First, to our knowledge, there is no clear sign that "world-cities" have exceptional growth rates in a local/national context. For example, the largest cities in the US, which are very well connected internationally, currently experience *smaller* growth rates than medium sized cities in the South and Southwest, which have less significant international exchanges.

Second, it seems, additionally, that the system of world-cities follows different rules in terms of their rank-size distribution (for example, see Jiang, Yin, and Liu¹, or Luckstead and Devadoss²), which might signal that they should not be regarded as an urban system in the traditional sense quantified in the manuscript.

Does the issue of scale (i.e. the physical size of the country) is also considered in terms of intervening opportunities (in addition to the gravitation constant)? See: Simini F, Gonzalez MC, Maritan A and Barabasi A-L (2012) A universal model for mobility and migration patterns. Nature 484: 96–100. doi: 10.1038/nature10856 PMID: 22367540

The physical size of the urban system (distance) plays only an implicit role in our analysis, affecting no doubt the magnitude of migratory flows between specific pairs of places. However, because we take these flows from data, we don't explicitly use the Gravity law, and thus also not the gravitational constant nor a specific distance function.

The emphasis of our analysis is the bilinear structure of the flows between two places and their dependence of each place's population. In this sense, any distance dependence for the flows will produce the results of the paper. Specifically, we use a generalized version for the flow currents – compatible with the gravity law, or indeed with the radiation model of the paper suggested – that is split up into two parts, a symmetric (s_{ij}) and anti-symmetric (δ_{ij}) flow. For the results in the manuscript, the anti-symmetric part plays the crucial role because it is responsible for cities changing their rank in the urban hierarchy and therefore for the convergence to any long-time relative city size distribution.

Conversely, the symmetric part – regardless of how strong the flows are – is associated with a demographic equilibrium, such as Zipf's law. Thus, as we point out in the manuscript, the observation – or assumption (!) – of symmetric flows in migration (as is the common form of the gravity law) is associated with a stationary distribution of city sizes. A (small) breakdown of this symmetry is typical of most real urban systems and has a meaning as information. We added some comments to the manuscript to clarify these issues.

Reviewer 2

To derive the Zipf's law, the authors assume that the rate of change, ϵ , in equation

¹Jiang, Bin, Junjun Yin, and Qingling Liu. "Zipf's law for all the natural cities around the world." *International Journal of Geographical Information Science* 29, no. 3 (2015): 498-522.

²Luckstead, Jeff, and Stephen Devadoss. "Do the world's largest cities follow Zipf's and Gibrat's laws?." *Economics Letters* 125, no. 2 (2014): 182-186.

(12), has the time-average going to zero. How can such an assumption be justified? The authors talk about some cities growing faster than others during certain periods in history, and then slower at later or earlier times, but to me this doesn't sound like a good argument for this mathematical assumption. Further, ϵ depends on x , which is a solution of the equation. Can we even make such an assumption logically? (I suppose one could assume this, solve the equation, and then show that the assumption was true, but I don't think this has been done here, please clarify if I am wrong). If the fact that the mean of $\epsilon \rightarrow 0$ as $t \rightarrow \infty$ is an assumption of the model, then what happens to the results when this assumption is relaxed? (It appears to be a crucial assumption for the analysis).

The condition $\epsilon \rightarrow 0$ for long times is not an assumption we take in the model itself, but a necessary condition to explain Zipf's law starting with basic demographic dynamics. Systems that do not share this property will not show a Zipfian structure of relative city sizes, as they contain certain "[...] preferences associated with birth, deaths, and migration.". This is one of the key points we are trying to make later in the paper when we introduce how deviations can be measured in terms of information. (See Methods for details). We have added some comments to the paper to make this clarify this question.

The population of people has been growing exponentially on the time scales of interest. How is this compatible with the assumption of the growth rates averaging out to zero?

Everything in the model is indeed generally growing exponentially ! It is only **relative** growth rates (for x not N) that must be zero on long time averages for Zipf's law to emerge. The mathematics in the model make this clear, hopefully. The deviation of relative growth rates – when the overall exponential growth of the system is subtracted – has a fundamental meaning in terms of selection and information in population dynamics, which has interesting implications for cities. We attempted to clarify this point in the revised manuscript.

The authors mention using data at several points in the paper (e.g. to justify their assumptions, to estimate the various scales, etc) – but it is not really explained how the data were used. It would be important to make these procedures more transparent.

Thank you. We have now added an additional Section to the Methods, which clarifies how and which data we used.

The authors talk about numerous deviations from Zipf's law that have been reported. On page 16 they state: "It is only on the average over many structure vectors at long times that Zipf's law emerges as a good characterization of the city size distribution." This appears to be the central result of the paper: even though Zipf's law doesn't really hold at any particular moment of time, it holds on average. In order to relate the theoretical findings to the subject of the paper, it would be necessary to find evidence of this in real data. The authors use data to validate parts of their model – can these same data (at different time points) be used to illustrate that the Zipf's law really holds if we average things out in time?

This is an excellent point! Thank you. We have now added an analysis of US historical data starting with the first Census of 1790 that establishes this behavior.

Because the US has had decennial census where the population size of cities has been counted since 1790, authors such as Gibson³ compiled the sizes of the top 100 cities every 10 years until 1990. Peakbagger.com made an attempt to extend this data to estimate the size of the largest 20 cities in terms of their *functional* extent between 1790 to 2010⁴.

We used the data from Gibson to create Figure 4, which depicts the size-rank distribution of the structure vector for every decade (dashed lines), as well as the cumulative time average over all structure vectors (red). Zipf's law is also shown (blue dashed) for comparison. The inset depicts the deviation of the structure vectors in terms of the

³<https://www.census.gov/population/www/documentation/twps0027/twps0027.html>

⁴<https://www.peakbagger.com/pbgeog/histmetropop.aspx>

Kullback-Leibler divergence from Zipf's law (blue) for each decade and from the the average from 1790 until the specific year (red). We observe – as stated – that, after over a few decades, the temporal average is always closer to Zipf's Law than the city size distribution in any single year. In the recent years, the US city size distribution began diverging from Zipf's law, so that the cumulative temporal average also shows some growing deviations but to a much lesser extent.

The analysis of only the top 20 urban areas based on Peakbagger's data is similar but we chose not to include it in the paper because these authors express some reservation on the consistency of the data. The figure is shown here below for completeness.

We believe that this analysis answers the question asked by the Reviewer and makes the paper stronger.

Reviewer 1

[...] that the intention of the authors is not to deepen the real demographic processes that make up urban growth, but rather to engage in an exercise full of mathematical virtuosity (I do not have the skills required to judge its accuracy) to derive a form of distribution of city sizes, at the cost of simplifying assumptions that add nothing to our knowledge of the cities, and which also do not allow to hope to bring closer the proposed model of empirical observations made on these processes. [...] The choice of the term "environment" seems to me excessively questionable because it can lead readers interested in the cities toward other interpretations.

We respectfully disagree with the Reviewer. We believe that our work adds to the fundamental understanding of how urban systems work and how quantitative empirical patterns of relative city sizes in geography can be explained by their inescapable demographic dynamics.

Zipf's law is a mathematical statement and requires mathematical theory and methods as a basis for its explanation. We fundamentally agree with the Reviewer that papers should be about ideas, and it is for this reason that we relegated the more difficult mathematical derivations to the Methods section, which functions as a supporting appendix.

We believe the arguments and derivations in the manuscript will appeal to the readers of Nature Communications, as the topic is of interest for readers with different backgrounds. The manuscript connects well known mathematical results in formal demography and population biology (ergodic theorems), financial mathematics (multiplicative random growth), quantitative geography (laws of geography) and information theory in a way that provides mutual consistency and insights between these fields.

Reviewer 3 agrees with our assessment and states that "The topic of size distributions is of interest to a wide community of scholars from various disciplines.". Reviewer 2 additionally comments: "I think this is a very interesting paper that studies an interesting question by methods that come from statistical physics and applied math."

We hope that on a second reading and in light of the comments below, the Reviewer can

appreciate the connections between geography, demography and population biology that the paper brings forth and the value of Zipf's law and its exceptions in this broader and better grounded formalism.

In their model presented page 4, the authors without any discussion nor empirical evidence make an important assumption "international migration to city i , as well as to and from non-urban places, can be incorporated into the vital rates". Knowing the important role of migrations, either international in the case of US, or from rural areas towards cities at some historical periods of their development in all countries of the world, this can hardly be accepted.

Thank you for this question. Expressing international migration from outside the city in this way is not an assumption we make, it is rather a way to mathematically express those dynamics and be able to include them in the model. Given information about other specific places (e. g. specific international cities) they can be readily included in the matrix \mathbf{A} . In this way, we do not assume that an urban system is restricted by political borders. Indeed the results of the paper show that an urban system is any set of cities with strong mixing, meaning strong mutual migration flows between them.

Naturally, political boundaries introduce restrictions to people's movement. For US urban areas the proportion of population change due to international migration is very small, about an order of magnitude lower than natural rates and internal migration. Thus, urban systems might often only be plausible within a nation or a region such as the European Union, with free circulation of people. We have adapted the manuscript to clarify these issues.

The trick is obvious in this passage on page 9 where the register of communication which passed smoothly equations of quantum physics with breaking symmetry to the ranking of size of the cities may perhaps appeal to some followers of physics theoretical but does not spare the understanding of the specialists of the cities [...]: "There are some analogies to Eq. 6 from other systems that may provide us with some additional intuition. The dynamics of Eq. 6 is analogous to a Schrödinger Equation in imaginary time, with Hamiltonian $H = A - I$, where I is the identity matrix. Such dynamics are dissipative, converging to a zero temperature solution determined by the ground state of the interaction Hamiltonian H . The dominant eigenvector of this matrix plays the role of this ground state to which the dynamics converges to at long times. This "ground state" breaks the symmetries of the initial state and determines in principle any specific city size rank ordering resulting from the interaction structure between cities, m_{ij} " [...] Another example on page 14 when the introduction of stochasticity in the growth model is commented in these terms: "This represents a flow of probability across the urban hierarchy and is somewhat analogous to the solution of the heat equation on a conducting one-dimensional bar with boundary conditions at different temperatures, implying a flow of energy through the medium."

Arguments of symmetry and ground states and similarities between dynamical models are common across not only physics, but population biology, computer science and demography. Pointing out these formal similarities provides many readers (such as the other two Reviewers) with points of familiarity, which guides them towards clearer interpretations and mathematical solutions they know from other contexts. These correspondences do not mean that systems are the same, merely that they are analogous in certain analytical regimes.

Establishing these formal connections is an essential point of developing theory and addressing an interdisciplinary audience, especially the wide variety of readers from Nature Communications. For these reasons, and asking the Reviewers for their indulgence, we decided to keep the analogies in the manuscript.

One can understand and approve the deliberate undertaking of authors to found a physics of urban systems that is universal (in the sense that it takes as little as possible account of historical and social evolution) through the transfer of established models for the physical sciences.

We are sorry that the Reviewer interpreted the paper in this way: This is not at all what

the paper is doing! The fundamental "universal" theory here is demography: the basic accounting of births, deaths and migration in each city.

Moreover, our main point is that the "historical and social evolution" of each place – and associated people's life choices – matter! This is why the "universality" of Zipf's law should not be expected in general: deviations carry information, which means precisely non-random choices that speak of people's agency.

We also show that it is only by averaging out all these social and historical factors that Zipf's law may emerge (if it does...) as a sort of "universal" pattern. Indeed, the manuscript provides tools to trace back such deviations to specific events, and, additionally, it can quantify the impact those events had on the structure of the urban system and how durable they are.

In other words, we hope that the Reviewer sees the paper indeed as pointing to the importance of historical and social factors – interpreted through the lens of demographic change, as it must be to predict cities' populations – in shaping the actual real distribution of city sizes. We have made some change to reflect these points.

I think the concluding sentence that best reveals the contradictory nature of this posture is: " Seeing Zipf's law and other laws of geography in this light helps us appreciate the information content of associated deviations in real urban systems, and trace them back to specific choices and preferences associated with births, deaths, and migration. The approach developed here for urban systems relies only on the analysis of the transition probability between types against a general background of multiplicative growth". Which of course brings nothing more to the geographer who deals with these questions. That is sad, because geographers for instance have demonstrated in other studies the role of a series of factors explaining major variations in the shapes of the observed distributions of city sizes that are much less vague than the invocation of "specific choices and preferences associated with births, deaths, and migration".

We do agree on this point. In this passage, we are commenting on previous models being too simple (and causing the abstractions that the Reviewer dislikes) and the need to go back to demography – which is inescapable - and the fundamental choices made by people on where to live, die and move.

We do believe that our work provides important mathematical methods and new insights into the dynamics behind Zipf's law and the meaning of other structural patterns in geography. By starting with demography, we show rigorously that urban systems in general will almost never have a structure that perfectly resembles Zipf's law. However, in the absence of specific preferences for places over the long term, the average structure of relative city sizes will tend towards approximating Zipf's law.

As can be currently observed in the US according to Census data and depicted in Fig. 4 in the manuscript, after WWII, the US urban system started to deviate from Zipf's law, after approximating it very closely. (Curiously this cusp was the time when Zipf made his statistical observations.)

The proposed explanation and methods introduced in the manuscript make it possible to pin down the point in time when systems start to deviate from any expected pattern and allow closer investigations into what factors might cause such deviations. The strong, long lasting, and increasing deviations now observed might signal, for example, that the US urban system is in a transitory state with strong preferences for certain specific places such as the mid-sized cities of Texas and the Southwest. However, such statements for specific urban systems need further context-specific investigation, for which the manuscript proposes helpful tools.

We hope that the Reviewer will appreciate some of these points that we tried to better explain in the revised manuscript.

Again, we reiterate our gratitude for all Reviewers constructive comments, which helped us substantially improve the manuscript.

REVIEWER COMMENTS

Reviewer #1 (Remarks to the Author):

This new version of the article is a little easier to read. Social science readers, demographers and urban systems specialists will not find much usable material in it and will remain baffled by the reference to "individual choices in terms of births, deaths and migration" line 326 as the ultimate explanation of urban growth. But the authors' aim is probably not to reach this audience. The model developed by the authors has the merit of staging population flows between cities to generate a variety of city size distributions that more or less conform to Zipf's law. In the discussion, I remain unconvinced by the statement given in lines 259-262 "We have shown that the demographic dynamics of urban systems can result in different city size distribution and how Zipf's law, the gravity law and other coarse-grained statistical regularities in geography emerge approximately from these dynamics as averages when certain symmetries are restored over sufficiently long times". Could the authors recall here under which conditions a "gravity law" "emerges" from their model? and could they give one or two examples of the other "coarse-grained statistical regularities in geography" that emerge from these dynamics?

Details:

Reference 8 is not accurate (already mentioned in my first evaluation) the right name of the author is François Moriconi-Ebrard

Reviewer #3 (Remarks to the Author):

I have now read the revised MS and the comments to my review and I feel my questions have been answered.

Thus, I recommend the paper be published.

As for the additional references I suggested, I suggest the author re-consider adding at least:

1. L. Benguigui, M. Marinov. arXiv preprint arXiv:1507.03408 (2015).
2. L. Benguigui, M. Marinov. arXiv preprint arXiv:1607.00856 (2016).

as these papers provide a thorough review of the size distributions of natural and social systems.

The authors' decision on this issue, however, does not change my recommendation regarding the publication of this work.

This work has its roots in demographic dynamics, whose tools and concepts were laid out by Caswell. The ms connects well known results in different disciplines (geography, demography, population dynamics, statistical physics) and has the merit to investigate why people detect deviations from Zipf's law. They find that in general urban systems should almost never have a structure that resembles Zipf's law. Birth, deaths and immigrations are sufficient to account for such deviations, but are also important for understanding why Zipf's law represents an attractor in the space of city distributions. I also think that this is of interest to a wide community of scholars.

Although I think the ms provides interesting insights into the dynamics of Zipf's law and other structural patterns in urban systems, sometimes I have found (ex., p.9) the jargon too heavily connected to the statistical physics community. This has the advantage to engage people (like me), but it has the disadvantage to give the impression that the authors are only translating some models from one field into another, like an exercise with no much depth. Also, it does not convey to geographers or specialists in city distributions the real importance of their achievement. For instance, rotational invariance is important for physicists, applied mathematicians and engineers, but it is not clear what that implies for geographers, urbanists, economists and demographers. Using language borrowed from some fields does not necessarily brings in new insights into another unrelated field. Thus I would certainly recommend a change in the overall jargon, especially some parts.

There are some other "more or less" technical aspects, which I do not fully understand:

- on p.11 the authors claim that they introduce "a slow non-linearity in each city's growth rate". What is a "slow" non-linearity? maybe "small"? other?
- it seems that the authors assume that there are no correlations between N_i and N_j hence $\langle N_i N_j \rangle = \langle N_i \rangle \langle N_j \rangle$, which I think they should somehow justify, especially when they introduce the stochastic formulation. These correlations could introduce non-trivial effects.
- it is not entirely clear to me how stochasticity is introduced in eq.13. Is \bar{v} a random variable or all the other terms as well? It seems to me that here stochasticity is introduced in a rather naïve way. Given that $\bar{\delta}_i$, v_i depend on x_i , why is eq. 13 linear in x_i (I mean the noise term does not depend on x_i)?
- in fig.2 there is a time evolution which implies a change in the rank of cities with time. However the authors do not seem to say much about whether the time scales of such evolution is in agreement with empirical data. Does the model predict the correct scales of temporal evolution on which cities reach a Zipfian behaviour?

In summary, I think the paper provides interesting insights into the general mechanisms of city formation and evolution, but I also find that the jargon sometimes obscures the implications for geography. They should also provide us with some implications which we should expect on the basis of the deviations that they find around Zipf's law. For example, it would be interesting to know something about the average extinction time $\langle t(k) \rangle$ of a city with a given rank k . Is it something like $\langle t(k) \rangle = t_{max} k^{-w}$? This may have interesting implications for small cities.

Reply to Reviewer's Comments

Reviewer 1

[...] will remain baffled by the reference to "individual choices in terms of births, deaths and migration" line 326 as the ultimate explanation of urban growth.

We thank the Reviewer for this comment, as this is a point that we hope to make crystal clear in the paper. Because the laws of geography are purely demographic and apply to the entire population (not segmented by types), the only processes that generate dynamics are indeed "[...] *specific* choices in terms of births, deaths and migration".

We don't think this is controversial: it is just (demographic) accounting. The key word in the Reviewer's comments, in this light, is "ultimate". We do not offer ultimate causes why people have children, die and move in various different places; this is very difficult to do with any level of accuracy. What the paper shows is that demographic dynamics channel a lot of the complexity that goes into these life-defining choices and influences in terms of just a few aggregate quantities (births, deaths and migration across a system of cities).

We made some changes to the text to make this distinction clearer.

Could the authors recall here under which conditions a "gravity law" "emerges" from their model? and could they give one or two examples of the other "coarse-grained statistical regularities in geography" that emerge from these dynamics?

Thank you, we want to emphasize that the starting equation (Eq. 2) is basic demography and does not constitute a modeling choice. Modeling assumptions come in when we explore explicit forms for the migration currents, which we show lead to different city size distributions.

Eq. 8 parameterizes the migration currents in a form that shows how the gravity law (in terms of its population dependence) can emerge from the dynamics. Specifically, we decompose it into a symmetric part (where the number of people migrating in each period between cities $i \leftrightarrow j$ is the same in both directions) and an antisymmetric part (this is always possible mathematically for a quantity with origin-destination indices). We then show in Eq. 9 how the standard form of the gravity law emerges from the symmetric part only, and the meaning of its parameters.

Later in the paper, Eqs. 11-13, show that the vanishing of the anti-symmetric part of the migration current happens naturally under the average over cities and over time, so that the gravity law "emerges" under this "coarse-graining" (averaging), even if it is not a feature of specific flows between any two cities at any particular time.

On time scales over which the gravity law emerges (as an average feature of flows), and if the vital rates (birth rates minus death rates) are also city size independent (mean and variance), one recovers Gibrat's law, Eqs. 9-13, because these quantities together determine a city's population growth rate. And if Gibrat's law holds under sufficient

long time averaging, the rest of the paper shows how Zipf's law emerges, see Eqs. 14-15 and surrounding text.

To the Reviewer's question, the other two examples of "coarse grained regularities" are Gibrat's and Zipf's laws. The point of the paper is to show the sequence of conditions necessary for these "laws" to emerge, starting with basic demography. And to show that one does get different results when at least some of these conditions - which, we agree with the Reviewer, may sometimes be too restrictive - fail to apply.

We hope these comments are clarifying. We have made some changes to improve the presentation.

Reviewer 4

Although I think the ms provides interesting insights into the dynamics of Zipf's law and other structural patterns in urban systems, sometimes I have found (ex., p.9) the jargon too heavily connected to the statistical physics community. [...] it is not clear what that implies for geographers, urbanists, economists and demographers. Using language borrowed from some fields does not necessarily bring in new insights into another unrelated field. Thus I would certainly recommend a change in the overall jargon, especially some parts.

In light of these comments and comments of Reviewer 1 and 2, we have changed the language in several parts of the manuscript and hope that the presentation makes the formal arguments more accessible for a broader readership.

on p.11 the authors claim that they introduce "a slow non-linearity in each city's growth rate". What is a "slow" non-linearity? maybe "small"? other?

Thank you for this question. Technically, we believe that the statement is correct: it means that the antisymmetric part of the migration current $\bar{\delta}_i$ is an average over the structure vector $x_i = \frac{N_i}{N_T}$. We use the term "slow", because this quantity is a temporal rate, so its "smallness" corresponds to a long time effect in the dynamics of the x_i .

Thus, short term observations of the system (x_i) will not reveal the non-linearities, as they only become observable over longer times.

This "slowness" effect is visualized when Figures 1a and 2a are compared. Even though the initial environment and the initial population structure are the same, the non-linear case (Figure 2a) takes a few orders of magnitude longer to reach a numerically stable state.

Thus, we think the use of the term "slow non-linearity" is appropriate and intuitive, but we have made changes to the presentation to make this clearer: thank you.

It seems that the authors assume that there are no correlations between N_i and N_j hence $\langle N_i N_j \rangle = \langle N_i \rangle \langle N_j \rangle$, which I think they should somehow justify, especially when they introduce the stochastic formulation. These correlations could introduce non-trivial effects.

Thank you for calling this point out, on which we have indeed not been sufficiently clear. In writing Eq. 9, we extract a factor of $\frac{N_i N_j}{N_T}$ out of the migration current, but introduce the two "intensive" quantities s_{ij} and δ_{ij} , (the symmetric and anti-symmetric parts of the current). This can always be done, but transfers the correlations between i, j to s_{ij}, δ_{ij} .

How these correlations are handled is different for the two parts: The symmetric part plays no role in the dynamics of the relative size of cities, no assumptions are made about its form at all. The anti-symmetric part plays the critical role, but enters the dynamics only via the population average $\bar{\delta}_i = \sum_{j=1}^{N_c} \delta_{ij} x_j$. All correlations between N_i and N_j present in the anti-symmetric component are included, but only contribute to the dynamics via this aggregate sum.

We have made some changes in the text to make these points clearer, but also tried to

keep the discussion not too technical.

It is not entirely clear to me how stochasticity is introduced in eq.13. Is \bar{v} a random variable or all the other terms as well? It seems to me that here stochasticity is introduced in a rather naïve way. Given that δ_i , v_i depend on x_i , why is eq. 13 linear in x_i (I mean the noise term does not depend on x_i)?

This was the point of the discussion in your previous question. The "slow-nonlinearity" in $\bar{\delta}_i$ is still there and is part of the numerical solutions. If the dependence of the quantities that go into ϵ_i on population size is strong and persistent one still can get asymptotically stable city size distributions (by the ergodic theorems), but they will not be Zipf's law! The treatment of the quantities that go into ϵ_i as stochastic is *not* necessary. The paper at this point simply shows that if these intensive quantities fluctuate in time (which they do empirically) and *if* their averaged behavior over long times behaves as a simple stochastic process (and only then) one gets Zipf's law (following essentially Gabaix's argument¹).

We do agree with the Reviewer that the last conditions may appear too strong (or naïve) at face value. But - as the paper shows - it is the fact that they may become much more natural under averaging (because of limit theorems) that may lead to approximate "coarse-grained" laws of geography.

Our objective is to show what (perhaps naïve) conditions are necessary. We intentionally leave their plausibility to the reader. This also sets up better posed empirical questions for future studies.

in fig.2 there is a time evolution which implies a change in the rank of cities with time. However the authors do not seem to say much about whether the time scales of such evolution is in agreement with empirical data. Does the model predict the correct scales of temporal evolution on which cities reach a Zipfian behaviour?

This is a good question. Depending on the nature of the dynamics, we defined two time scales t_* (page 9) and t_r (page 16). We also did have some commentary later in the text in this direction, namely:

page 9: "Using the values of migration flows in the US urban system over the last decade, we can compute the value of t_* . It is rather long - of the order of several centuries- when measured using census data and a little shorter using data on tax returns.

page 16: "This leads to an effective symmetry of the relative city size distribution on the time scale $t_r = \frac{1}{2\sigma^2}$, when growth rate fluctuations vanish. It can be estimated using US data to be typically very long, on the range of many centuries or even millennia."

It is this long characteristic time-scales that create difficulties in answering the Reviewer's question empirically.

To do so we have done two things: We have verified that the time scale t_* characterizes the convergence time in the simpler matrix dynamics numerically. For the approach to Zipf's law, we can get t_r from the analytic solution in the Methods section. We have also included Figure 4, which shows some approach to Zipf's law over more than two centuries for the US in order to test t_r .

We have added some text to further clarify these points. Thank you.

They should also provide us with some implications which we should expect on the basis of the deviations that they find around Zipf's law. For example, it would be interesting to know something about the average extinction time $\langle t(k) \rangle$ of a city with a given rank k . Is it something like $\langle t(k) \rangle = t_{\max} k^{-w}$? This may have interesting implications for small cities.

Thank you for this question. We have extended the discussion with additional possible

¹Gabaix, Xavier. "Zipf's Law and the Growth of Cities." American Economic Review 89.2 (1999): 129-132.

implications this work might have for the understanding of urban systems.

In addition, we are happy to provide the answer to the behavior of the "extinction time", which is implicit in the analytic solutions in Methods.

As Eq. 17 shows, there is only one time scale in the dynamics that approach Zipf's law: It is $t_r = \frac{1}{2\sigma^2}$. The rank-dependent decay time to fall through the lower size boundary can be read out from the various solutions (Eqs. 26-33), which express different initial conditions for the conditional probability $P[x, t; x_0, t_0]$ for a city to have relative size x at a later time t , given that it started with size x_0 at time t_0 . We define the time interval $\Delta t = t - t_0$.

The "extinction time" $\langle \Delta t(k) \rangle$ is defined implicitly through the conditional probability $P[x_m, t; x_0, t_0] = p \sim 1$, where x_m is the minimum size, which close to Zipf corresponds to maximal rank $k_{\max} \sim 1/x_m$. Observation of the exact solutions in Methods tells us that the leading time is $\langle \Delta t_r(k) \rangle \sim \frac{1}{2\sigma^2} \ln \frac{x}{x_m} = \frac{1}{2\sigma^2} \ln \frac{k_{\max}}{k}$. (There are subleading terms that depend on the exact initial condition.)

This shows that the natural dependence of the extinction time on rank is *logarithmic*, not power law. This is clear in the analysis in Methods, since only the variable $y = \ln \frac{x}{x_0}$ enters the solution. The longest extinction time belongs naturally to the largest city (with initial $k = 1$): it is $\langle \Delta t_r(k = 1) \rangle \sim \frac{1}{2\sigma^2} \ln k_{\max}$, which depends how large the urban hierarchy is (how many cities in the urban system since $k_{\max} = N_c$).

We thank the Reviewer again, and have introduced this result in the Methods section.

Again, we reiterate our gratitude for all Reviewers constructive comments, which helped us substantially improve the manuscript.

REVIEWERS' COMMENTS:

Reviewer #4 (Remarks to the Author):

The authors have changed the language in several parts of the manuscript, the presentation now seems a bit more accessible.

The stochasticity is introduced in a simplistic way, but anyway the paper shows that if some intensive quantities (δ_i) fluctuate in time (and they do empirically) and if their averaged behaviour over long times behaves as a Gaussian white noise process, then it is possible to get the Zipf's law. deviations to this are also shown.

As for the correct time scales of the evolution of the model, the authors have identified two scales and they have added a commentary in the main text. Though long characteristic time-scales pose some difficulties from the empirical standpoint, they have tried to clarify these points in the ms. The reader has the opportunity to assess the plausibility.

They have finally introduced the extinction time as a function of rank and, interestingly, they find that this is logarithmic, not power law.

In summary, I found these clarifications satisfactory and useful to the overall understanding of the paper.